# Mitotic Catastrophe Induced in HeLa Tumor Cells by Photodynamic Therapy with Methyl-aminolevulinate

**DOI:** 10.3390/ijms20051229

**Published:** 2019-03-11

**Authors:** Marta Mascaraque, Pablo Delgado-Wicke, Alejandra Damián, Silvia Rocío Lucena, Elisa Carrasco, Ángeles Juarranz

**Affiliations:** 1Departamento de Biología, Universidad Autónoma de Madrid, 28049 Madrid, Spain; marta.mascaraque@uam.es (M.M.); pablo.delgado@uam.es (P.D.-W.); alejandra.damianv@gmail.com (A.D.); silvialucenablas@gmail.com (S.R.L.); carrascoelisa@gmail.com (E.C.); 2Instituto Ramón y Cajal de Investigaciones Sanitarias, IRYCIS, 28034 Madrid, Spain

**Keywords:** photodynamic therapy, HeLa tumor cells, cell death, mitotic catastrophe, spindle elements

## Abstract

Photodynamic therapy (PDT) constitutes a cancer treatment modality based on the administration of a photosensitizer, which accumulates in tumor cells. The subsequent irradiation of the tumoral area triggers the formation of reactive oxygen species responsible for cancer cell death. One of the compounds approved in clinical practice is methyl-aminolevulinate (MAL), a protoporphyrin IX (PpIX) precursor intermediate of heme synthesis. We have identified the mitotic catastrophe (MC) process after MAL-PDT in HeLa human carcinoma cells. The fluorescence microscopy revealed that PpIX was located mainly at plasma membrane and lysosomes of HeLa cells, although some fluorescence was also detected at endoplasmic reticulum and Golgi apparatus. Cell blockage at metaphase-anaphase transition was observed 24 h after PDT by phase contrast microscopy and flow cytometry. Mitotic apparatus components evaluation by immunofluorescence and Western blot indicated: multipolar spindles and disorganized chromosomes in the equatorial plate accompanied with dispersion of centromeres and alterations in aurora kinase proteins. The mitotic blockage induced by MAL-PDT resembled that induced by two compounds used in chemotherapy, taxol and nocodazole, both targeting microtubules. The alterations in tumoral cells provided evidence of MC induced by MAL-PDT, resolving mainly by apoptosis, directly or through the formation of multinucleate cells.

## 1. Introduction

Photodynamic therapy (PDT) is a minimally invasive treatment modality. Although still emerging, it is already a successful therapeutic procedure used for the management of oncologic and non-oncologic diseases [1,2]. Within the oncologic disorders, PDT presents an essential advantage due to the minimal side effects produced during and after treatment, through the specific actuation against malignant cells [3]. PDT is approved in clinics for the treatment of several forms of non-melanoma skin cancer [4] and also for lung and oesophageal neoplasias, among other type of cancers [5,6]. The therapy is based on the combination of three essential components: light, oxygen and a photosensitizer (PS). None of them are individually toxic, but their combination triggers the photodynamic reaction which is tumor ablative by generating highly reactive oxygen species (ROS), mainly singlet oxygen (^1^O_2_) [6]. These molecules have short half-life and a destruction radius in the range of nanometers, being, therefore, the main targets the organelles where PSs are located [7,8]. The activation of the PS is obtained by irradiating with light of appropriate wavelength, generally in the red region of the visible spectrum (620–690 nm); with high capacity to deeply penetrate into the tissues [9]. The events initiated after PDT administration can lead to tumor cell death directly or indirectly by causing damage to tumor microvasculature and inducing a local inflammatory reaction [10]. More than a dozen compounds have been clinically approved or are in clinical trials for their use in PDT. Among them, there are the 5-aminolevulinic acid (ALA) and its ester derivatives, such as the methyl-aminolevulinate (MAL). Both are prodrugs commonly used in the clinical practice that are enzymatically turned into Protoporphyrin IX (PpIX), a potent PS [11].

It has been described that PDT can lead to the three main cell death pathways: apoptosis, necrosis and autophagy. The mode of cell death induced by this therapy depends on the type of tumor cell, the treatment doses and particularly the intracellular localization of the PS, which is related to the damage to specific organelle [12,13]. Different cell components can be targeted by PDT, including plasma membrane, lysosomes, mitochondria, Golgi apparatus, endoplasmic reticulum (ER), nuclei and cytoskeleton [14]. For instance, if the PS is localized in mitochondria PDT drives malignant cells mainly to apoptosis [15]. Otherwise, necrosis is the major cell death modality induced by PDT when compounds are localized in the plasma membrane, since treatment promotes then membrane permeability followed by the breaking down of cell components [16,17]. It has also been described that PDT irradiation of certain PSs, as the ER-localizing PS hypericin, can lead to cell autophagy [18]. This cell fate occurs when cell-repairing attempts in response to damage fail, triggering apoptosis in cancer cells [19].

The mitotic catastrophe (MC) had been previously defined as a type of cell death triggered by aberrant mitosis and executed either during mitosis or in the subsequent interphase [20]. According to the Nomenclature Committee on Cell Death, MC can constitute an oncosuppressive process that culminates in apoptotic, necrotic or senescence-mediated elimination of mitosis-deficient and genomically unstable cells [21]. Among the distinct therapeutic cellular targets that could drive to MC, cytoskeleton elements exhibit great relevance [22]. Targeting the cytoskeleton is an important goal for anticancer therapies since it plays significant roles in processes related to tumor progression such as cell mobility, division and vesicular transport [23]. In this sense, well-known chemotherapeutic components impeding microtubules (MTs) dynamics have been widely used in anticancer therapies, such as alkaloids, nocodazole (Nc) or toxoids, like taxol (Tx) [24,25]. In addition, there are several published works describing that MTs are affected by PDT by some PSs [14,24,25,26]. Specific PSs have been related to MC induction. Such treatments were able to induce cell cycle blockage at the metaphase-anaphase transition, which leads to MC followed directly by apoptosis and/or after the formation of giant cells [26,27].

Although the use of the metabolite MAL is clinically extended, the mechanisms underlying cancer cells death after MAL-PDT are not completely characterized. Here we present evidences indicating that MAL-PDT leads to MC in HeLa cells by affecting MTs dynamics and the expression and location of proteins implicated in mitotic regulation.

## 2. Results

### 2.1. Cell Toxicity

The PDT treatment combining MAL and red-light irradiation (635 nm) in carcinoma HeLa cells induced variable cell toxicity, determined by the MTT (3-(4,5-dimethylthiazol-2-yl)-2,5-diphenyltetrazolium bromide) assay. The toxicity was dependent on both; MAL concentration (0.3 or 1 mM) and light dose (2.25–6.75 J/cm^2^) (Figure 1). In contrast, cell viability of HaCaT cells, used as non-tumorigenic control cells, after the same treatments was always higher than 80%, which supports the selectivity of the treatment. The incubation of HeLa and HaCaT with MAL (0.3 or 1 mM), in the absence of light, did not induce any significant toxicity. In the same way, no toxic effects were detected in the cells after red-light irradiation alone with the highest light dose used in this work (6.75 J/cm^2^) (Table 1). Based on these results, we have established specific conditions as sublethal dose in HeLa cells for the rest of experiments (MAL 0.3 mM and red light doses of 2.25 J/cm^2^).

### 2.2. Subcellular Localization and Production of PpIX

The intracellular localization of the PS, PpIX, produced from exogenous MAL, was evaluated by fluorescence microscopy under UV (365–390 nm) or green (510–550 nm) exciting light (Figure 2a,b). Control cells (without MAL) did not show red signal (Figure 2a). However, HeLa cells exhibited a red emission due to PpIX production after incubation with MAL (0.3 mM) for 5 or 24 h. PpIX was located mainly at the plasma membrane (asterisks), although it was also observed diffusely in the cytoplasm and in some organelles (arrows), typically in a juxtanuclear position (Figure 2a). To determine the organelles in which PpIX was located, next we performed co-localization experiments with well-known biomarkers: for mitochondria (DIOC_6_), lysosomes (LTG), endoplasmic reticulum (ER-BW) and Golgi apparatus (CellLight Golgi-GFP) (Figure 2b). Images taken at different excitation wavelengths, confirmed the localization observed when cells were incubated only with MAL and thus indicates that the positive juxtanuclear region corresponds mainly to lysosomes, and to a lesser extent, to the ER and Golgi apparatus.

Since we detected changes in the cellular response to PDT when using different treatment conditions, we analyzed by flow cytometry the levels of PpIX produced in HeLa cells (Figure 2c). The production of PpIX after 5 h of incubation with MAL resulted to be dependent on the MAL concentration (0.3 vs 1 mM), whereas no significant differences were found due to the incubation times (5 vs. 24 h) at each MAL concentration (Figure 2d). In contrast, PpIX production in HaCaT cells was independent of both MAL concentrations and incubation times in all the experimental conditions tested (Appendix A). These results showed that HeLa cells produced higher levels of PpIX after 5 h of incubation with 1 mM of MAL in comparison with 0.3 mM.

### 2.3. Alterations in Cellular and Nuclear Morphology Triggered by PDT

General and nuclear morphology was studied in the HeLa cell line after MAL-PDT with sublethal dose (0.3 mM MAL and 2.25 J/cm^2^ red light), using phase contrast and fluorescence microscopy after staining with Höechst-33258 (Figure 3). Untreated HeLa cells presented a polygonal keratinocyte structure. The incubation with MAL or red light alone did not induce DNA damage (Appendix A); whereas 5 h after PDT, the cells showed a slight cellular retraction and many rounded mitotic cells could be observed (not shown). After 24 h of MAL-PDT, cell cultures presented a high number of cells with division-characteristic morphologies (mainly metaphases, normal and abnormal with chromosome fragmentation), which indicates arrest in mitosis induced by the treatment (Appendix A, control cells; and Appendix A, MAL-PDT cells). After 48 h of PDT, cells appeared with multinucleate and apoptotic morphologies (cell rounding, blebbling and shrink cells with vesicles all over the cell surface and chromatin fragmentation) [28] (Figure 3a,b).

Cell cultures treated with the sublethal dose were analyzed by flow cytometry after labeling with propidium iodide (PI). Figure 3c shows the cell cycle distribution outlines and the percentages of cells in each cycle phase, comparing control cells with 24 and 48 h after PDT. Control cells presented a typical outline, with the G0-G1 frequency three times higher than G2-M, and low proportion of both, cell death and polyploidy. It can be noticed that 24 and 48 h after PDT there was a sharp decline of G0-G1 frequency, while there was an increase of G2-M. It was also observed an increment on the percentage of polyploidy cells (approximately from 2% to 7%) 48 h after PDT. Finally, 48 h after PDT, the percentage of dead cells increased to 7%. The synthesis phase was maintained stable over time. Together with the phase contrast images, all these data indicate a blockage in mitosis in response to PDT, with a subsequent activation of cell death mechanisms, while not rounded cells continued growing normally.

These studies were complemented with analyses by fluorescence microscopy after staining unfixed cells (floating and attached) with ethidium bromide (EB) and acridine orange (AO). After 24 h of MAL-PDT with the sublethal dose, these assays allowed us to distinguish viable, dividing and multinucleated cells all fluorescing in green, as well as dead/apoptotic cell morphologies, fluorescing in red (Figure 4a).

In order to investigate the mechanism by which PDT caused mitotic blockage in HeLa cells, we treated the cells in parallel with two known anti-cancer drugs, whose target are MTs: Tx and Nc. Treatment of HeLa cells with these compounds (0.05 µM Tx or 0.25 µM Nc) caused the well-known mitotic arrest and characteristic nuclei morphologies, which were quite similar to those described after the application of MAL-PDT (Figure 4b). The number of cells with the morphologies described after different treatments and the EB-AO staining is shown in Figure 4c. PDT caused a significant increase of 47.5% of rounded dividing cells 24 h after treatment with respect to the control. At 48 h since after the treatment, the percentage of rounded cells was halved, and that of viable and multinucleated cells increased significantly. Tx and Nc treated cultures displayed an elevated percentage of cells in mitosis (65% and 39%, respectively) 24 h after treatment. After 48 h, as in PDT cells, the percentages of dividing cells also decreased, while apoptotic and multinucleated cells were more frequent.

### 2.4. MAL-PDT Induces Microtubule Alterations in the HeLa Line

Cytoskeleton components are involved, among other physiopathological functions, in cell proliferation as well as metastasis. Among them, MTs are implicated in the formation of the mitotic spindle in dividing cells [29,30]. The evaluation of MTs alterations caused by MAL-PDT was carried out by indirect immunofluorescence (IF) for α-tubulin 5, 24 and 48 h after treatment. Control cells in interphase showed a well-developed microtubular network formed from the juxtanuclear organizing region (Figure 5a) and cells in division exhibited well-organized bipolar mitotic spindles with the chromosomes aligned on the equatorial plate (Figure 5b). At 5 h after PDT, cells in interphase showed retraction accompanied with some MT disorganization (Appendix A). The most important changes were observed 24 h after MAL-PDT increasing the mitotic index (MI) compared to untreated control cultures as previously indicated. In addition, 85% of dividing cells were undergoing metaphase 24 h after PDT while in control cells equivalent to 25%. The majority of the mitotic spindles presented an abnormal morphology (multipolar and aberrant bipolar figures) being 87% of total metaphase after 24 h PDT (Figure 5a). Figure 5b shows representative images of normal and abnormal metaphases, differentiating between multipolar (Figure 5b-1) and aberrant bipolar with altered disposition chromosome on the equatorial plate (Figure 5b-2). The number of cells in division, also with altered spindles, 48 h after PDT was still high, and multinucleated and giant cells appeared corresponding to cells that have entered in G1 (Appendix A).

The results obtained after MAL-PDT regarding MTs organization/disposition were compared to those induced by the treatment with Tx or Nc. After 24 h of Tx treatment, the majority of dividing cells showed multipolar spindles and chromosomal fragmentation; Nc induced mostly bipolar spindles with disorganized chromosomes in the equatorial plate (Figure 5b). The altered disposition of the chromosomes was confirmed by IF for α-tubulin and anti-centromere antibodies (ACA) (Figure 5b). These data, along with the morphological assays, suggest that MAL-PDT induces MC in HeLa tumoral cells by targeting MTs.

### 2.5. Localization and Expression of Proteins Involved in Cell Division after MAL-PDT

In order to approach the cellular/molecular mechanism involved in the mitotic blockage and MC process, the expression pattern of different proteins implicated in these processes was evaluated by indirect IF and Western blot (WB). The first two molecules analyzed were aurora kinase A and B (AURK A and AURK B). AURK A collaborates in the regulation of entry into mitosis, helpful in centrosome maturation [31]. In HeLa control cells AURK A was located around the centrosomes and pericentriolar material in prophase; in prometaphase and metaphase on both sides of the poles according to the duplication of the centrosomes, and during anaphase was localized predominantly around the polar MTs, although a scattered signal was found in the middle body (Appendix A). In the case of cells treated with PDT, Tx or Nc, a more intense AURK A signal was visualized at the centrosome level, either with localization similar to that of the control, or disperse, with multiple signals, in multipolar cells (Figure 6a). This result coincides with that observed by IF for α-tubulin in cells with multiple poles.

AURK B, related to the regulation of anaphase output, centromere regulating correct chromosome alignment and segregation during mitosis [31]. In control cells was located in the centromeres in metaphase, in middle body and in the equatorial plate in anaphase and in telophase, cytokinesis and early G1 it was located in the dividing furrow (Appendix A). In cells treated with PDT, in general, a similar location to control HeLa cells was observed, except in those cells with an aberrant morphology of the mitotic spindle, as in case of tripolar cells, which presented AURK B in the middle body. The cells in mitotic arrest treated with Tx and Nc showed a scattered signal of AURK B, coinciding with the non-alignment of the chromosomes in the equatorial plate (Figure 6b).

In relation to this, an increase in cyclin B1 was observed 24 h after treatments, which correlates with the increase shown in dividing cells (G2/M) (Figure 6c). Cyclin B1 has a crucial function in cell division and its degradation is essential for exiting mitosis by activation of the anaphase-promoting complex (APC) [32].

Finally, due to the important role of γH_2_A.X in MC, we assessed its expression in HeLa cells 24 h after treatment (Figure 6d). As expected, control cells in interphase did not show expression of γH_2_A.X, only cells in division exhibited expression of the protein, necessary for the correct condensation of DNA during division. After MAL-PDT, both cells blocked in division as well as multinucleated cells positively expressed the protein. The percentage of positive control cells for γH_2_A.X was 12% whereas in treated cultures was 65%.

### 2.6. Mechanism of Cell Death

The general morphology of dead cells (observed by phase contrast microscopy), of nuclei (observed by fluorescence microscopy after Höechst-33258 staining) and the sub G1 phase (detected by flow cytometry) (see Figure 3) allowed us to postulate that MC induced by the sublethal PDT dose applied to HeLa cancer cells was mainly resolved by apoptosis. To confirm this hypothesis, HeLa cells were incubated with Z-VAD-FMK (broad-spectrum caspase inhibitor) before MAL-PDT. The results obtained showed that the caspase inhibitor protects the cells against apoptosis inducing an increase in cell survival (Figure 7a). Finally, we analyzed the expression of caspase 2 and 3; the first one implicated in activating MC and the second one in general apoptosis. Both caspases increased their expression consecutively, 24 and 48 h after PDT (Figure 7b). All these results indicate that the mechanism of cell death involved the participation of caspase and considering the results obtained by the morphology analysis, we concluded that MC was being resolved by apoptosis.

## 3. Discussion

The present in vitro study provides evidences of induction of MC in HeLa cells after MAL-PDT through MTs disruption and deregulation of the expression of key cell cycle proteins. Based on the specific accumulation of the PS in the tumor cells and subsequent irradiation with visible light, PDT enables maintenance of healthy cells while malignant cells are eradicated [1,2,3]. It has been broadly described that the main cell death pathways induced by PDT are apoptosis, necrosis and autophagy. However, less is known about the processes leading to them via MC [12]. In particular, nothing has been described about MAL-PDT as potential inductor of MC. This process had been described as a cell death mechanism related with defective mitosis regulation that leads to cell cycle arrest, mitotic spindle abnormalities, multipolar division, altered chromosome arrangement, aneuploidy and, finally, cell death [33]. Much of the treatments triggering this cellular response target cytoskeleton components, MTs in particular, or induce DNA damage [24,25]. Up to now, a few PSs have been clinically approved, including MAL, precursor of the endogenous PS PpIX, which presents a huge relevance in the topical treatment of non-melanoma skin cancer and the precancerous lesion actinic keratosis [34,35]. Despite of being currently a common treatment, cases of resistance or recurrence have been described [36,37]. Therefore, deeper knowledge of cellular mechanisms underlying cancer cell death is needed in order to prevent negative responses to PDT and to improve this therapy.

In this sense, by using HaCaT as a non-tumorigenic control cell line and the widely studied carcinoma cell line HeLa, we first evaluated the efficacy of PDT. Neither the light nor the prodrug individually administrated induced cell death. In addition, while HeLa cells were affected by PDT in a red-light (2.25–6.75 J/cm^2^) and MAL (0.3 or 1 mM) dose-dependent manner, HaCaT cells remained almost unaltered. These results confirmed PDT selectivity destroying cancer cells [1,2,3,6]. From the obtained results, we selected specific conditions to perform the rest of the experiments that induced a sublethal toxicity (lower than 50%) in HeLa population: 0.3 mM MAL, 5 h of incubation and 2.25 J/cm^2^ of red-light dose.

It has been established that the cell death pathway followed by cancer cells after PDT is directly related to, among others, subcellular location of the PS employed [12,15]. In that context, the studies performed on subcellular localization of PpIX in HeLa cells after MAL incubation indicated that the PS was accumulated mainly in plasma membrane and in cytoplasm and no differences were observed after the MAL incubation times evaluated (5 and 24 h). In addition, our co-localization studies showed that MAL induced PpIX were also located at lysosomes and Golgi apparatus level, without ruling out the possible presence of PS in the ER. These results are in disagreement with those obtained in oesophageal carcinoma cells, where ALA-induced PpIX appeared in the mitochondria [38]. Chen, R. et al., reported that ALA-induced PpIX location in DHL (follicular lymphoma) cells was detected mainly in ER and mitochondria, but also in lower levels in the lysosomes [39]. From all these results, it is tempting to consider that once PpIX is produced in the mitochondria, it can reach to other organelles, in our case, mainly the cell membrane, cytoplasm, lysosomes and Golgi apparatus.

In addition to PS intracellular location and ROS-generated in that specific area, PpIX intracellular concentration has also shown great relevance in PDT effectiveness. In our case, PpIX accumulation increased with MAL concentration, which was consistent with the decrease in cell viability found with the MAL dose used in this work. These results also support previously published data indicating that an increase in PpIX production is related to an increase in the induction of cell death tumor cells being directly associated with a higher ROS production [40].

Morphological evaluation of PDT HeLa cell cultures showed cell retraction a few hours after the treatment and cell rounding, being more evident 24 and 48 h later. This result suggests that MAL-PDT was inducing mitotic arrest at metaphase-anaphase transition. In fact, 24 h after MAL-PDT we detected a MI of 40%, in contrast with only 5% in control cells. In addition, most of the cells in division were in metaphase and exhibited abnormal MTs morphologies. After 48 h, the number of giant cells with macronuclei and multinucleated cells increased, revealing an increment in endoreplication. We also determined an increase in cells with apoptotic features. These morphological observations were confirmed by flow cytometry, reflecting an abnormal distribution of the treated cultures among the cell cycle phases both 24 and 48 h after MAL-PDT. A sudden decline of the G0-G1 frequency was appreciated, followed by an increase in G2-M at both times points evaluated after the treatment. Furthermore, 48 h after PDT there was an increment in the proportion of multinucleated cells and dead cells. These data suggest that after accumulation in G2-M, some cells could skip the blockage and continue their cycle giving rise to a higher number of aneuploidies and greater genomic instability which could lead, as a consequence, to more aggressive and resistant tumors [33,41].

All the aforementioned results support that cells were blocked at mitosis with a subsequent activation of cell death processes that could be explained by MC induction and resolution. This mechanism can occur after different situations of cell damage: due to MTs impairment, problems affecting the mitotic machinery sensed during the M phase or chromosomal defects [26,33]. Accordingly, MTs due to their role in cell division constitute a fundamental target for antineoplastic treatments, such as Tx and Nc [30,42]. Tx is a MT stabilizing agent that binds with high affinity to polymerized tubulin. It promotes the growth of these cytoskeletal elements and inhibits their disassembly, blocking the cell cycle in the transition from metaphase to anaphase, inducing finally apoptosis [43,44]. Cells in the presence of Tx cause an activation of APC, complex that promote the anaphase, that favors the reduction of cyclin B1 levels and, therefore, the early exit of mitosis by slippage [44]. On the other hand, Nc is a MT destabilizing agent that has been observed leading to cell blockage in the metaphase-anaphase transition [45]. In this sense, after MAL-PDT the same effects of those induced by Tx and Nc treatments in HeLa cells were observed: mitotic blockage, spindle abnormalities, disorganization of chromosomes, multipolar cell division, and generation of daughter cells with macronuclei or multinucleated.

In addition, the assessment of protein expression by IF or WB revealed modifications in molecules related to cell division after PDT treatment. AURK A exhibited a very similar location and expression pattern in great number of treated cells than in the control, but was increased in cells with two or more centrosomes, indicating a potential increase in the AURK A synthesis [46,47] as a consequence of centrosome amplification in response as the ROS produced after PDT, as it has been previously described [48]. In addition, AURK B and ACA were dispersed in their location indicating absence of alignment of the chromosomes in the equatorial plate at metaphase. All these results support that MTs are an essential target of PDT with MAL, as occurs with other PSs [14,24,25,26].

Another fundamental protein in cell division is cyclin B1. In this sense, an increase in the expression levels of this protein accompanied with mitosis blockage have been described after Nc treatment [49]. In line with previous work by other authors, this study has shown an increase in the expression of cyclin B1 24 h after PDT, Tx and Nc treatments.

In keeping with this, double strand DNA breaks (DSB) in mammalian chromosomes lead to S139 phosphorylation of the H_2_AX histone, also called gamma-H_2_AX (γH_2_A.X). This phosphorylated histone is required as a checkpoint for cell cycle arrest and DNA repair against DSB [50]. Therefore, the increase in the expression of γH_2_A.X observed after PDT indicates DNA damage like a factor of MC.

All these results confirm that the application of sublethal doses of PDT in HeLa cells induces MC, as it has also been described in other studies using different cell lines and PSs [26,27], indicating that the effects caused by them are similar to those of the anti-carcinogenic agents Tx and Nc [42,43,44,45].

Finally, MC can be resolved in different ways: (a) cells directly die without exiting mitosis; (b) reach the G1 phase of the subsequent cell cycle (through a phenomenon that is known as mitotic slippage, which generates giant cells and then dies); (c) exit mitosis and undergo senescence [21]. In this study, we have observed an increase in multinucleated and apoptotic morphologies 48 h after PDT, which was prevented by the caspase inhibitor Z-VAD-FMK. In addition, our results indicate an increase in the expression of caspase-2, followed by the increment of caspase-3 expression and the condensation of chromatin 24 and 48 h after PDT, indicating that apoptosis via MC was happening as it has been reported in other studies with other PS compounds [51,52].

In summary, the present study shows that MAL-PDT in HeLa cells induce MC, altering both the dynamics of MTs and chromosome arrangement. MC was finally resolved by slippage or apoptosis. A greater knowledge of the response related to tumoral cell death mechanism triggered after MAL-PDT could contribute to optimize this therapy to avoid potential tumor recurrences.

## 4. Materials and Methods

### 4.1. Cell Culture

For the in vitro studies, we used HaCaT cells, a spontaneously transformed but non-tumorigenic human keratinocytes cell line (Cell Line Service, Eppelheim, Germany) and the human HeLa carcinoma cell line (obtained from ATCC), used as reference in vitro control in many laboratories focused on PDT action. Cells were routinely grown as a monolayer in F25 flasks (Fisher) or culture dishes with or without glass coverslips (Menzel-Gläser) placed inside the dishes, using Dulbecco’s modified Eagle’s medium (DMEM) supplemented with 10% (*v*/*v*) fetal bovine serum (FBS), 50 units/mL penicillin, and 50 μg/mL streptomycin (all from Gibco, Paisley, UK). Cell cultures were performed in an incubator with 5% of CO2 at 37 °C and 95% humidity. Treatments were performed when cultures reached a 60–70% confluence.

### 4.2. Reagents and Antibodies

Methyl-aminolevulinate (MAL), Taxol (Tx) and Nocodazole (NC) were obtained from Sigma-Aldrich (St. Louis, MO, USA). The broad-spectrum caspase inhibitor benzylocarbonyl-Val-Ala-Asp-fluoromethyl ketone (Z-VAD-fmk) was purchased by BD Biosciences (Villeurbanne, France). MTT (3-(4,5-dimethylthiazol-2-yl)-2,5-diphenyltetrazolium bromide) (Sigma-Aldrich, St Louis, MO, USA). The following primary antibodies were used: mouse anti α-tubulin (Sigma-Aldrich), mouse anti AURKA (Abcam, Boston, MA, USA), mouse anti β-actin (Sigma-Aldrich), rabbit anti AURKB (Abcam), rabbit anti P-H_2_A.X (S139) (Cell Signaling Technology, Danvers, MA, USA), rabbit anti caspase 3 (NeoMarkers, Fremont, CA USA); mouse anti caspase 2 (BD Biosciences, Villeurbanne, France), mouse anti cyclin B1 (Biosource, Nivelles, Belgium) and Human ACA-Texas Red (Antibodies Incorporated, Davis, CA, USA). The secondary antibodies used were: mouse IgG-Alexa 488 and rabbit IgG-Alexa 546 (Invitrogen, Carlsbad, CA, USA), mouse IgG Peroxidase and rabbit IgG Peroxidase (Thermo Scientific, Rockford, IL, USA). Fluorescent markers were: 3’ dihexyloxocarbocyanine iodide (DIOC6) for mitochondria, Lysotracker Green (LTG) for lysosomes, ER-Tracker Blue White-DPX for endoplasmic reticulum and CellLight Golgi-GFP marker for Golgi apparatus (all of them from Invitrogen, Carlsbad, CA, USA). Höechst-33258 (Riedel-de Haën, Hannover, Germany), acridine orange (AO) (BDH, Poole, UK) and ethidium bromide (EB) (Sigma-Aldrich) were used for cellular staining.

### 4.3. Treatments

For PDT, MAL was prepared at an initial concentration of 10 mM in deionized sterile water. For phototreatments, cells were cultured in plates of 24 wells and incubated 5 h with appropriated MAL concentrations (0.3 and 1 mM) in DMEM culture medium without FBS. Afterwards, cells were irradiated with variable light doses (2.25 to 6.75 J/cm^2^) by using a red-light emitting diode source (WP7143 SURC/E Kingsbright) with an irradiation intensity of 6.2 mW/cm^2^ (as measured by Coherent Lasermate powermeter) and a emission peak at λ = 634 ± 20 nm. To minimize light refraction, cells were irradiated from the bottom of the plates (Appendix A). After irradiation, the medium containing MAL was changed by DMEM with 10% of FBS. We have observed that the presence or absence of MAL during irradiation did not affect HeLa cell viability (evaluated by the MTT assay, see below) (Appendix A). Therefore, from these observations, cell irradiation was performed in the presence of MAL. Time-lapse microscopy was used to evaluate the mitotic blockage process by taking images each 30 min for a total of 24 h using a digital Leica inverted microscope DMI 6000 B. In order to evaluate the implication of caspase enzymes in the cell death process induced by PDT, cells were incubated with 25 μM (final concentration) of Z-VAD-fmk for 5 h in presence of MAL before light irradiation. The inhibitor binds to the catalytic site of caspases blocking apoptosis. After irradiation, the cells were incubated in complete medium for 24 h at 37 °C until evaluation. The experiments were repeated at least three times. For cellular drug-induced blockage, HeLa cells were treated with Tx that promotes MTs stabilization or Nc, a MT assembly inhibitor. When cultures reached 60% confluence, cells were treated for 24 or 48 h with 5x10^−8^ M of Tx and 2.5x10^−7^ M of Nc prepared in DMEM with 10% FBS [26].

### 4.4. Cellular Toxicity

Toxicity of the different concentrations of MAL alone or followed by red-light irradiation on HeLa and HaCaT cells was evaluated 24 h after treatments by the MTT assay. This method is widely accepted as a quantitative colorimetric assay for cell toxicity and it is based on active cell metabolism. After treatments, cells were incubated with MTT (at a final of 50 µg/mL) at 37 °C for 3 h, the culture medium was then removed, and the precipitated formazan was dissolved in DMSO. Absorption was measured at 542 nm in a spectrophotometer (Espectra Fluor 4, Tecan, Bradenton, FL, USA). Cellular toxicity was expressed as the percentage of formazan absorption from MAL-PDT cells compared to control cells (which received neither MAL nor PDT). To test the effect of light only on cell toxicity, we performed control experiments using only the red-light irradiation on cells 6.75 J/cm^2^, the highest dose used in this work. The results obtained are shown as mean values and standard deviations from three independent experiments.

### 4.5. Morphological Studies

Changes in cell morphology after PDT were analyzed using bright field illumination, or fluorescence microscopy after staining with Höechst-33258. At different times after treatments (24–48 h), floating (detached) cells in the culture medium were collected from each dish and centrifuged at 1000 rpm for 5 min, the supernatant was eliminated and cells were fixed with 50 µL 70% ethanol (−20 °C). Nuclei of detached cells were stained by adding 5 µL of Höechst-33258 (1 mg/mL). The type of cell death was determined analyzing nuclei morphology according to morphological criteria previously published [28]. The staining of the cells with AO-EB allowed distinguishing between viable and dead cells. After 24 or 48 h of the treatments (PDT, Tx or Nc), EB and AO were added to the cultures at a final concentration of 50 µg/mL. Immediately after EB-AO staining cells were observed by fluorescence microscope under blue excitation light. According to the fluorescence color observed, cells were classified as follows: viable and dead cells fluorescing in green or orange, respectively [53]. In addition, the AO-EB staining allowed distinguishing cells in mitosis, and with rounded morphology. At least 500 cells were counted for each treatment condition.

### 4.6. Subcellular Localization of MAL

Cells were grown on coverslips and when they reached around 70% of confluence, were incubated with MAL 0.3 mM either for 5 or 24 h at 37 °C. Then, cells were briefly washed with PBS (phosphate buffered saline), mounted on slides and observed in situ with a fluorescence microscope using UV excitation light. The subcellular localization was analyzed comparing the fluorescent signal pattern emitted by PpIX with that obtained by fluorescent markers of known selective subcellular accumulation: 3,3’ dihexyloxocarbocyanine iodide (DIOC6) for mitochondria, Lysotracker Green (LTG) for lysosomes, ER-Tracker Blue White-DPX for endoplasmic reticulum and CellLight Golgi-GFP marker for Golgi apparatus following the instructions given by the manufacturers. For that, cells were incubated with MAL 0.3 mM for 24 h, and the cells were further incubated for 5 min with the markers. For Golgi apparatus co-localization studies, cells were co-incubated with MAL and CellLight Golgi-GFP for 18 h. The co-localization assays were performed twice and evaluated using the Olympus fluorescence microscope using the appropriate filter sets. Quantification of percentage of co-localization was performed using FIJI software (available online: http://fiji.sc/Fiji), which is an open source image-processing package based on ImageJ software (National Institutes of Health, Bethesda, MD, USA). We first established, separately, the threshold for positive cells in PpIX and in the organelle marker fluorescent images, obtained under the corresponding exiting light: UV (360–390 nm), blue (450–490 nm) and green (510–550 nm). The positive PpIX area as well as that shared with the fluorescent organelle marker was measured. Then, the percentage of shared positivity with respect to the organelle marker was calculated. At least the fluorescence of 100 cells were measured with the corresponding exciting lights.

### 4.7. Determination of Intracellular Synthesis of PpIX

Cells grown in F25 were incubated with 0.3 or 1 mM MAL for 5 h or 24 h. Then, cells were washed with PBS, trypsinized and centrifuged for 10 min at 2000 rpm. After centrifugation, the supernatant was removed and cells were resuspended and fixed with formaldehyde 3.7% in PBS for 15 min at room temperature. Cellular suspensions were centrifuged, fixator was removed, cells were resuspended on PBS and stored at 4 °C on darkness between 12 h or 24 h until evaluation. PpIX emission measurement was made by flow cytometry (Cytomics FC500, Beckman Coulter, Corston, UK), with an excitation line of 620 nm and emission of 670 nm. Data were referred to the control (basal synthesis of PpIX). Each experiment was repeated three times.

### 4.8. Cell Cycle Evaluation

Cells distribution throughout the cell cycle phases was studied by flow cytometry. Culture flasks at 24–48 h after phototreatment (0.3 mM of MAL for 5 h and a light dose of 2.25 J/cm^2^) medium containing floating cells were separated and attached cells were trypsinized. Floating and attached cells were collected together and centrifuged at 2000 rpm for 10 min. Afterwards, medium was discharged and cells were fixed in cold 70% (*v*/*v*) ethanol (−20 °C) for 5 min. After RNase A digestion (0.1 mg of RNase A at 37 °C for 30 min), the cells were stained with 50 mg mL PI for 30 min before analysis with flow cytometer (Becton-Dickinson FAC Scalibur, San Jose, CA, USA). PpIX has been already photodegraded and therefore, it does not interfere with PI fluorescence in cell cycle analysis (Appendix A). Measurements were taken on a Beckman Coulter EPICS XL-MCL flow cytometer with an argon laser line at 488 nm complemented with appropriate filters. Cell cycle experiments were repeated at least three times.

### 4.9. Immunostaining

Cells grown on the coverslips were fixed in cold methanol (−20 °C) for 7 min, washed with distilled water three times and stored with 0.1% Triton X-100-PBS until processing. For immunostaining, cells were incubated with primary antibody for 1 h at 37 °C, inside a humid chamber. After washing with PBS, cells were incubated with the secondary antibody for 45 min at 37 °C. Coverslips were then washed with PBS and mounted with ProLong® with DAPI (1 µg/mL). Images were taken with a fluorescent microscopy Olympus BX610. Mitotic index (MI) was determined after IF for α-tubulin by counting cells in division (prophase, metaphase and anaphase-telophase) and metaphases were divided in normal and abnormal figures, depending on whether or not the spindle apparatus was altered; 700 cells were counted for MI determination. For measuring γH2AX positive cells, 500 cells per condition, in three independent samples.

### 4.10. Western Blots

For Western blot analysis, cells were lysed in RIPA buffer (150 mM NaCl, 1% Triton X-100, 1% deoxycholate, 0.1% SDS, 10 mM Tris-HCl pH 7.2, 5 mM EDTA), containing the appropriate concentration of Phosphatase Cocktail and Protease Inhibitor Cocktail (Sigma-Aldrich). Protein concentration was measured by the BCA Protein Assay Kit (Termo Scientific Pierce, Rockford, IL, USA). The proteins were electrophoresed and blotted on Immobilon-P PVDF membranes (Millipore Co., MA, USA). Membranes were blocked in PBS-tween 0.1% with 5% non-fat dried milk for 1 h at 25 °C and then incubated with the first antibody overnight at 4 °C. After washing with PBS-tween 0.1%, membranes were subjected to the peroxidase-conjugated secondary antibody and developed by chemiluminescence (ECL, Amersham Pharmacia Biotech, Little Chalfont, UK) employing the high definition system ChemiDocTR XRS+ (Bio-Rad Laboratories, Hercules, CA, USA). The bands corresponding to the different proteins were digitalized employing the Image Lab version 3.0.1 (Bio-Rad Laboratories). This assay was performed three times for each protein.

### 4.11. Optical Microscopy

Microscopic observations were carried out using an Olympus BX61 epifluorescence microscope equipped with filter sets for fluorescence microscopy: ultraviolet (exciting filter BP360-390), blue (exciting filter BP460-490), and green (exciting filter BP510-550). Photographs were obtained with the digital camera Olympus CCD DP70 and processed using the Adobe Photoshop CS5 extended version 12.0 software (Adobe Systems Inc., San Jose, CA, USA).

### 4.12. Statistical Analysis

Data are expressed as the mean value of at least three experiments ± standard deviations (SD). The statistical analysis was made with the SPSS 15.0.1 software (SPSS Inc., Chicago, IL, USA). The statistical significance was determined using *t* test and analysis of variance (ANOVA), and *p* < 0.05 was considered statistically significant.

## 5. Conclusions

The present study shows that MAL-PDT in the human HeLa carcinoma cells induces accumulation of mitotic cells. Additionally, it induces the appearance of spindle abnormalities, chromosome disaggregation, and multipolar cell division generation of daughter cells with macronuclei or multinucleated. All these features are indicative of MC, which occurs probably by targeting MT. MC is resolved mainly by subsequent apoptosis.

## Figures and Tables

**Figure 1 ijms-20-01229-f001:**
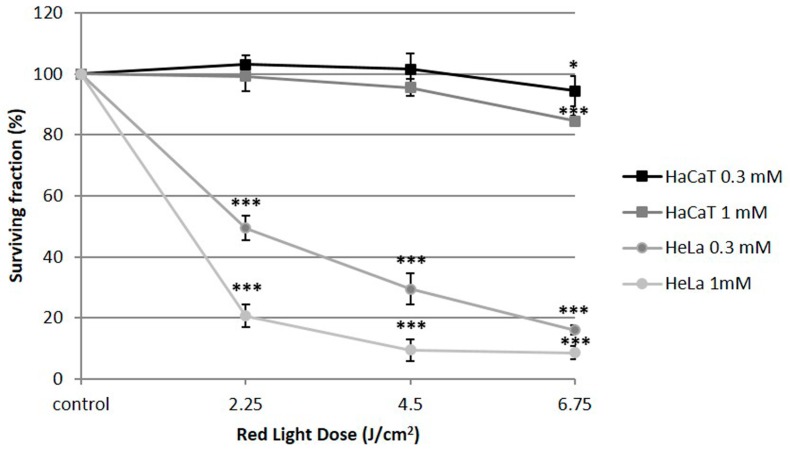
Phototoxicity induced by MAL-PDT (photodynamic therapy with methyl-aminolevulinate) in HaCaT (non-tumorigenic human keratinocytes cell line) and HeLa (cervical human carcinoma) cells. Cells were incubated with 0.3 or 1 mM of MAL for 5 h and then irradiated with red light at variable doses. In HeLa cells, the response to MAL-PDT was dependent of the concentration of MAL and the light dose. HaCaT were not sensitive to these conditions of PDT. The photoeffects were evaluated by the MTT test 24 h after treatments. Each value corresponds to the mean obtained from three independent experiments ± SD. (* *p* < 0.05; *** *p* < 0.001).

**Figure 2 ijms-20-01229-f002:**
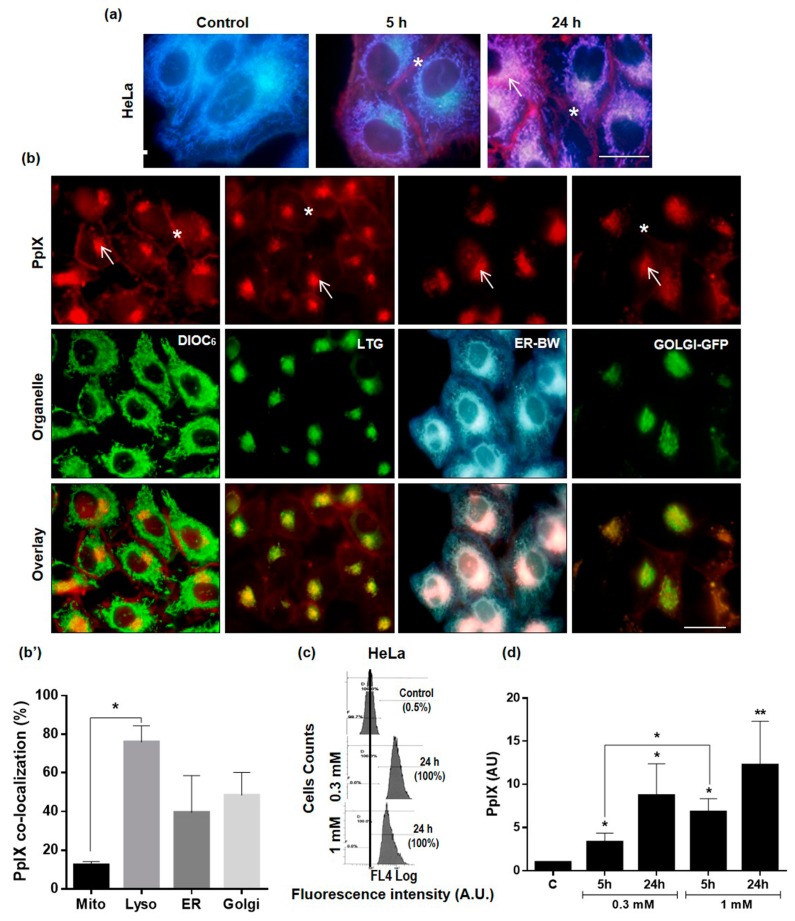
Subcellular localization and production of PpIX. Cells were incubated with MAL (5 and 24 h) and PpIX was observed in the fluorescence microscope under UV or green light irradiation. (**a**) Blue signal was mainly due to the autofluorescence of mitochondria, whereas the red signal, due to the PpIX production, was observed on the plasmatic membrane (asterisks) and diffuse in the cytoplasm (arrow). (**b**) Co-localization experiments of PpIX with different organelle markers. Cells were incubated for 24 h with 0.3 mM MAL and with fluorescent probes for mitochondria (DIOC_6_), lysosomes (LTG), endoplasmic reticulum (ER-BW), Golgi apparatus (Golgi-GFP). PpIX fluoresced in red whereas DIOC_6_, LTG and Golgi in green and ER in blue after green (510–550 nm), blue (450–490 nm) or UV (365-390 nm) exciting light, respectively. Arrow: organelle localization; asterisks: plasmatic membrane localization. (**b’**) Percentage of co-localization of PpIX with the different organelles quantified by using FIJI software. (**c**) Representative data obtained from flow-cytometry analysis of the PpIX production, after 0.3 or 1 mM MAL concentration, for 5 h of incubation. (**d**) PpIX production was relativized to its basal level in controls (not incubated with MAL). Each value corresponds to the mean obtained from three independent experiments ± SD. (* *p* < 0.05, ** *p* < 0.01). Scale bar: 20 µm.

**Figure 3 ijms-20-01229-f003:**
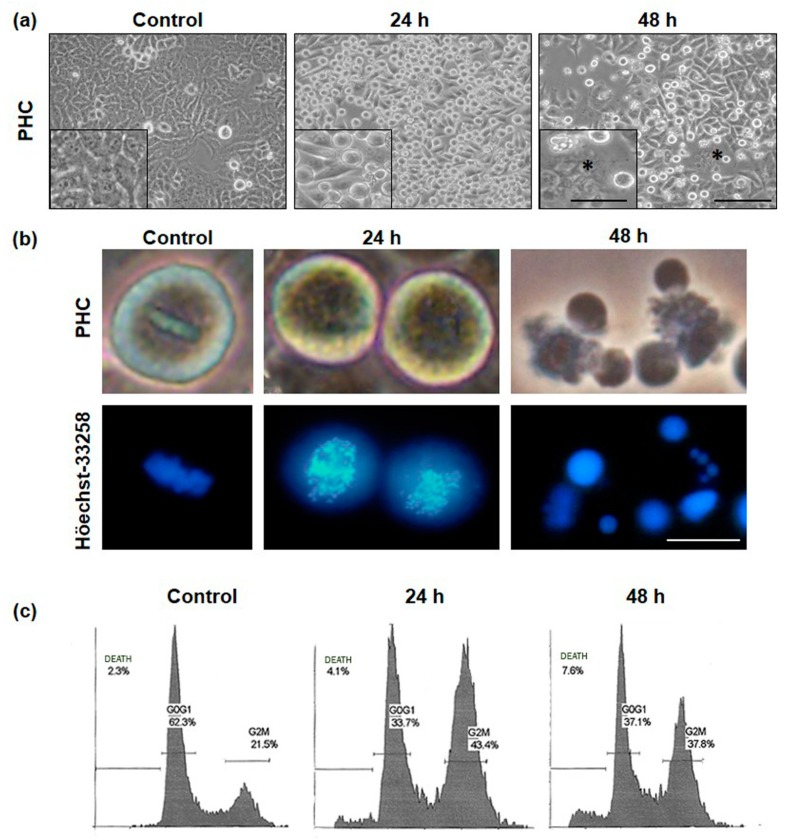
Cellular and nuclear morphology in control cells and after PDT (photodynamic therapy). Cells were observed by phase contrast microscopy (PHC). (**a**) Control HeLa cells presented an epithelial aspect; after 24 h treatment a high number of rounded mitotic cells could be seen in the cultures; after 48 h treatment, cells with multinucleated morphology appeared in the cultures (asterisk) and apoptotic morphologies. Scale bar: 100 µm; inserts 10 µm. (**b**) PHC and nuclei morphology observed by fluorescence microscopy after Höechst-33258 staining, after 24 h PDT mainly metaphases, normal and abnormal with chromosome fragmentation and after 48 h PDT apoptotic morphology. (**c**) Cell cycle distribution outlines in each cell cycle phase 0, 24 and 48 h after PDT. Scale bar: 20 µm.

**Figure 4 ijms-20-01229-f004:**
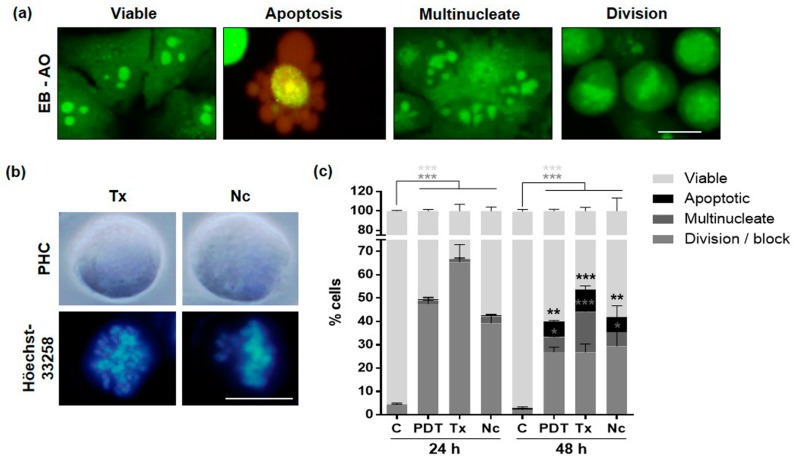
Morphology classification after fluorochome staining. (**a**) In vivo analysis 24 h after MAL-PDT by fluorescence microscopy after ethidium bromide (EB)-acridine orange (AO) staining. This staining allows differentiating between: (1) viable (fluorescing in green due to AO) (2) apoptotic (fluorescing in red/yellowish due to AO/EB), (3) multinucleate, and (4) dividing cells. (**b**) Cellular and nuclear morphology 24 h after treatment with 0.05 µM Tx or 0.25 µM Nc observed by PHC or Höechst-33258. (**c**) Percentage of viable, dividing, apoptotic and multinucleated cells 24 and 48 h after treatments application (values relative to control cells) (* *p* < 0.05; ** *p* < 0.01; *** *p* < 0.001). C: Control; Tx: taxol; Nc: nocodazole. Scale bar: 10 μm.

**Figure 5 ijms-20-01229-f005:**
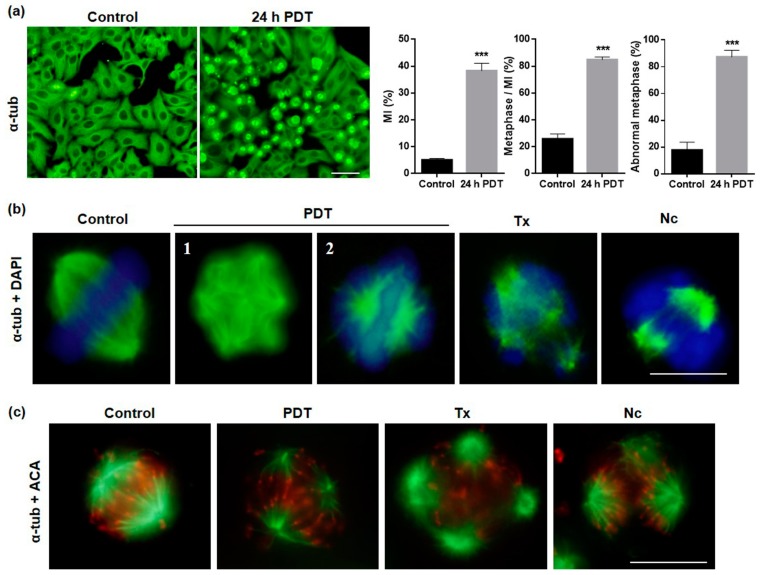
Microtubular and chromosomal alterations after PDT. Cells were evaluated 24 h after treatments by IF for α-tubulin. (**a**) Morphological changes in α-tubulin after 24 h PDT. Mitotic index, number of metaphases/MI and percentage of abnormal metaphases (*** *p* < 0.001). Scale bar: 50 μm. (**b**) Mitotic spindles of control and after MAL-PDT, Tx and Nc treated cells showing normal and altered morphology, respectively. DNA was stained with DAPI. (1) Multipolar cell, (2) aberrant bipolar with altered disposition chromosome on the equatorial plate. Scale bar: 10 μm. (**c**) Mitotic spindles (fluorescing in green) and centromeres (fluorescing in red) determined by IF for α-tubulin and ACA, respectively. Scale bar: 10 μm.

**Figure 6 ijms-20-01229-f006:**
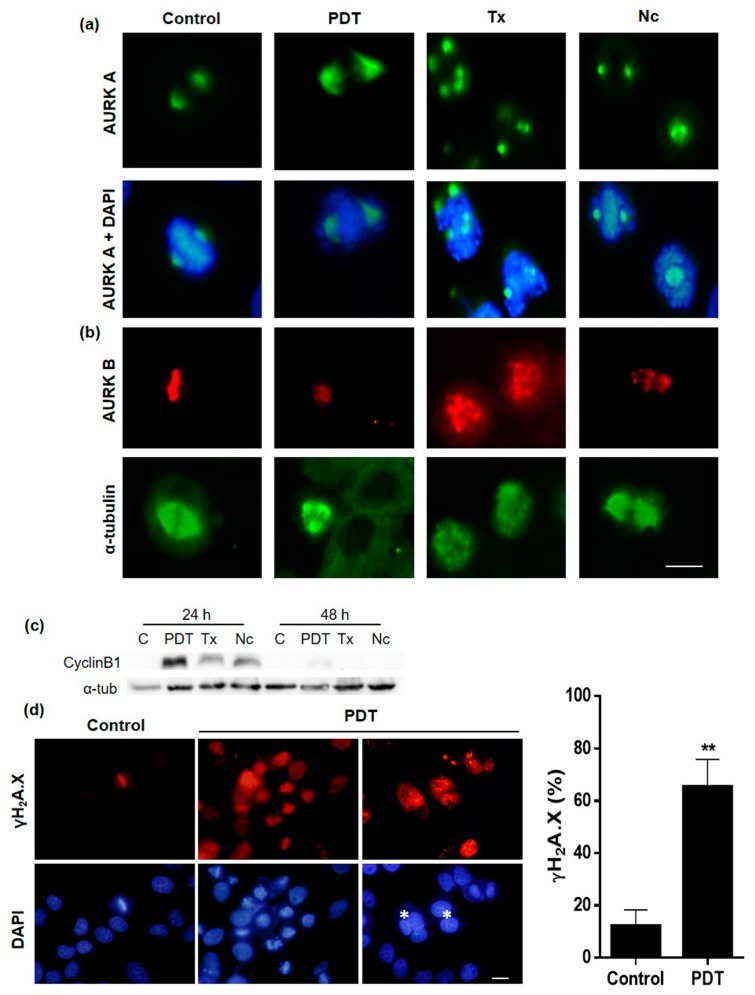
Localization and expression of proteins involved in cell division. (**a**) Localization of AURK A 24 h after treatment of PDT, Tx and Nc; the protein appears situated at the centrosome level. (**b**) Localization of AURK B after 24 h treatment of PDT, Tx and Nc; the protein was observed at chromosome level. (**c**) WB analyses of cyclin B1 and α-tubulin 24 and 48 h after the treatments. (**d**) Representation of positive HeLa cells by IF of γH_2_A.X in control and after PDT; asterisk: multinucleate cells. Scale bar: 10 μm. (** *p* < 0.01).

**Figure 7 ijms-20-01229-f007:**
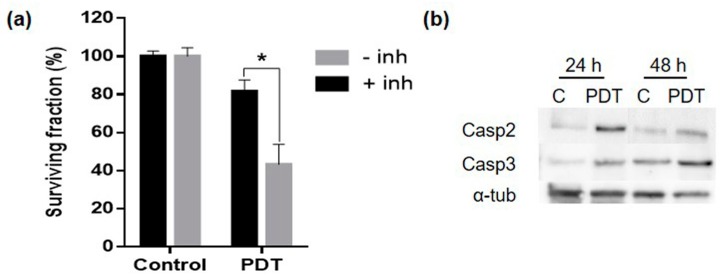
Mechanism of cell death. (**a**) Cell survival percentage after 24 h PDT with or without caspase inhibitor Z-VAD-FMK. (**b**) WB analysis of caspase 2, caspase 3 and α-tubulin 24 and 48 h after PDT. (* *p* < 0.05).

**Table 1 ijms-20-01229-t001:** Toxicity induced in HeLa and HaCaT cells by MAL or red light alone. Cells were incubated for 5 h with MAL at different concentrations or irradiated with the highest light dose used in the phototoxicity experiments. Toxicity was evaluated by the MTT test 24 h after treatments. Each value corresponds to the mean obtained from three independent experiments ± SD.

MAL Concentration (mM)	Light Dose (J/cm^2^)	HeLa	HaCaT
-	-	98.9 ± 1.10	100 ± 1.95
0.3	-	97.4 ± 2.30	104 ± 0.01
1	-	104 ± 5.17	96.88 ± 8.16
-	6.75	99.4 ± 1.30	101.69 ± 3.68

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
