# Peer review of "Mitotic Catastrophe Induced in HeLa Tumor Cells by Photodynamic Therapy with Methyl-aminolevulinate"

_ijms, 2019, doi:10.3390/ijms20051229_

Round 1
Reviewer 1 Report
The authors have done a thorough study on MAL, a precursor of Pp IX, to understand its mechanism of action. The authors suggest that PpIX + light, induces mitotic catastrophe and use several methods to determine this. I think that there are still several things that need to be addressed/clarified prior to publication. However, I believe that the work is publishable and will be interesting to other readers.
For the MTT assay, it is unclear whether the cells were refreshed prior to irradiation. If they were not refreshed, how much do you think is cell death caused by ROS generated externally? If they were refreshed, please include this. How were the cells irradiated, were they in 96 well plates or small dishes? How did the whole volume of cells get irradiated? Why not use a dose-response curve?
In the fluorescence microscopy, why was UV and green light used? It seems that unless a certain set of filter sets were used, the PP IX would also absorb and emit light from UV excitation as well as green light. What is the %colocalisation of the subcellular staining and the Pp IX? I think these images would benefit from using a confocal fluorescence microscope so that colocalisation can be better determined. Why not use FIJI to process your images?
For the Flow cyt. it would be helpful to include what wavelengths were used.
Figure 3 is difficult to understand. What is the dose (light and MAL) used for Figure 3a/b? Assuming 0.3 and the lowest light dose, it seems unlikely that the cells at 24 h would recover and are likely dead, yet this does not correlate with your MTT assay? Nor does the live/dead Flow cytometry? What is the scale for the insets? It should be possible to get a phase contrast image, which you can overlay with your hoechst staining. As of now, we have to believe that what you are denoting with an * is what you say it is. Was the control taken 24 or 48 h after treatment?
Does PpIX interfere with PI? Where is the resistant cell data? What are they resistant to?
The resolution on Figure 4b Hoechst, needs to be improved. Also, it is really difficult to understand the differences in sizes between phase contrast and hoechst. It would be easier to understand if the images were on the same scale. For Figure 4c, how many cells were counted for each population?
Figures 5b and 5c are very nice. Did you use the same microscope for these images? If so, why are the resolutions of the other images, which are zoomed so poor?
In Figure 6B. why not include the overlay? The yH2AX staining in Figure 6D, is not necessarily what is expected. In most yH2AX staining that I have seen, it is focal and solely in the nucleus (Oncogene volume23, pages2825–2837 (12 April 2004). Here, this looks like there is a lot of background/non-specific staining in the cytoplasm. How many cells were counted?
Author Response
The authors have done a thorough study on MAL, a precursor of Pp IX, to understand its mechanism of action. The authors suggest that PpIX + light, induces mitotic catastrophe and use several methods to determine this. I think that there are still several things that need to be addressed/clarified prior to publication. However, I believe that the work is publishable and will be interesting to other readers.
Comment 1 For the MTT assay, it is unclear whether the cells were refreshed prior to irradiation. If they were not refreshed, how much do you think is cell death caused by ROS generated externally? If they were refreshed, please include this. How were the cells irradiated, were they in 96 well plates or small dishes? How did the whole volume of cells get irradiated? Why not use a dose-response curve?
Response. Cells were cultured in plates of 24 wells and they were irradiated immediately after 5 h of incubation with MAL in DMEM without FBS. To minimize light refraction, cells were irradiated from the bottom of the plates (Supplementary Figure 5A). After irradiation, the medium containing MAL, was changed by DMEM with 10% of FBS and cells were further incubated for 24 h before performing the MTT assay. This has been clarified in the M&M section of the MS (line 422-427). MAL is not by itself a photosensitizer compound; is the precursor of the PS PpIX, therefore, we did not expect that MAL irradiation could induce ROS. In fact, data previously obtained in our laboratory, not included in the preceding version of this MS, indicate that the presence or absence of MAL do not affect HeLa cell viability; these data are now included a Supplementary Figure 5B. The treatment conditions used in this work were selected in light of the survival data shown in Figure 1.
Comment 2 In the fluorescence microscopy, why was UV and green light used? It seems that unless a certain set of filter sets were used, the PPIX would also absorb and emit light from UV excitation as well as green light. What is the % colocalisation of the subcellular staining and the Pp IX? I think these images would benefit from using a confocal fluorescence microscope so that colocalisation can be better determined. Why not use FIJI to process your images?
Response. PpIX has several absorption bands: the main one is the Soret band located at 400 nm, and also presents Q bands at 500, 546 and 625 nm. These bands correspond with the filters of UV (exciting filter BP360-390) and the green exciting light (exciting filter BP510-550) used in the microscope. Under all those exciting lights PpIX fluoresces in red; UV exciting light allowed also to observe the mitochondria autofluorescence and the green exciting light was employed in the co-localization studies since none of the fluorescent probes for organelles used in this study could be excited with green light. According to the suggestion made by the referee, we have processed the co-localization images using the FIJI software to determine the percentage of co-localization of each fluorescent probe and PpIX and this has been included in the M&M section (line 475-481). For that, we first established, separately, the threshold for positive cells in PpIX and in the organelle marker fluorescent images, obtained under the corresponding exiting light: UV (360–390 nm), blue (450–490 nm) and green (510–550 nm). The positive area for PpIX as well as that shared with the fluorescent the organelle marker was measured. Then, the percentage of shared positivity with respect to the organelle marker was calculated. Two representative images are shown in the Word file. We have included a bar diagram with the obtained results (Figure 2B’) confirming that PpIX is mainly localized in lysosomes.
Comment 3 For the Flow cyt. it would be helpful to include what wavelengths were used.
Response. As it was indicated in the M&M section, PpIX measurements were obtained by using the Cytomics FC500 cytometer (λexc=620 nm and λem=670 nm).
Comment 4 Figure 3 is difficult to understand. What is the dose (light and MAL) used for Figure 3a/b? Assuming 0.3 and the lowest light dose, it seems unlikely that the cells at 24 h would recover and are likely dead, yet this does not correlate with your MTT assay? Nor does the live/dead Flow cytometry? What is the scale for the insets? It should be possible to get a phase contrast image, which you can overlay with your hoechst staining. As of now, we have to believe that what you are denoting with an * is what you say it is. Was the control taken 24 or 48 h after treatment?
Response. In relation to improve the figure 3 we have modified the text to clarify the results as follows (line 40): “… cell line after MAL-PDT with sublethal dose (0.3mM MAL and 2.25 J/cm2),…”.
The differences in cell death mentioned by the reviewer could be explained as follows. From the MTT assay we concluded that cell viability, which estimated metabolic activity of the cells, was altered in approximately the 50% of MAL-PDT treated cells. We cannot indicate that this decrease in the metabolic rate is strictly revealing cell death. That was the reason why we performed different assays in order to determine the cellular fate of these cells. Our hypothesis was that a great part of these 50% of non-viable cells was dead or undergoing the process of cell death, after PDT (that is why these conditions were named sublethal dose). We found that these cells, representing the 50% of the cells in the culture, correlated with the percentage of rounded (detached) cells. As a first attempt to determine how these cells were dying, we analyzed the cell cycle distribution after the sublethal dose, observing that cells were blocked in G2-M phase, despite the fact that the estimation of cell death was approximately 4%; that could be the reason of the reduction in metabolic activity detected by the MTT assay. This cell blockade lead in time mainly to cell death; only a few percentage of cells scape from the blockade giving multinuclear and giant cells (Supplementary Figure 3), so we performed a set of assays to underlie the evolution of these rounded cells.
We have added in the legend of figure 3: “Scale bar: 20 µm; insets 10 µm”.
According with the comment of the referee we have modified the figure 3B. Höechst fluorescence images have been changed and we have added their corresponding phase contrast images. We have modified the legend of the Figure 3B accordingly.
The asterisk indicates a multinucleate cell that can be seen at higher magnification in the inset. We have added the asterisk into the inset.
Yes, corresponding controls were performed at 24 and 48 h after treatments.
Comment 5 Does PpIX interfere with PI? Where is the resistant cell data? What are they resistant to?
Response. The absorption and emission peaks of PI are 493 and 636 nm, respectively, and those of PpIX are 500 and 631, respectively. Therefore, their signals could interfere if both compounds are present at the same time. Flow cytometry was performed 24 h after light exposure. At this time, PpIX has been already photodegraded (Tyrrell, J., Campbell, S., & Curnow, A. Protoporphyrin IX photobleaching during the light irradiation phase of standard dermatological methyl-aminolevulinate photodynamic therapy. Photodiagnosis and photodynamic therapy, 2010, 7, 232-238) and therefore, it does not interfere with PI fluorescence in cell cycle analysis. In any case, we have incorporated a new supplementary figure demonstrating PpIX photobleaching immediately and 24 h after PDT (Supplementary Figure 5C).
We appreciate the reviewer’s suggestion, so we have modified line 160: “…while not rounded cells continued growing normally.” With resistant cells, we are referring to those cells that do not suffer alterations after the MAL-PDT treatments.
Comment 6 The resolution on Figure 4b Hoechst, needs to be improved. Also, it is really difficult to understand the differences in sizes between phase contrast and hoechst. It would be easier to understand if the images were on the same scale. For Figure 4c, how many cells were counted for each population?
Response. According to the suggestion made by the referee, new Höechst micrographs at higher resolution, and their corresponding phase contrast images have been included in figure 4B. For the ethidium bromide / acridine orange assay 500 cells were counted per condition, in 3 independent samples. This is indicated in the M&M section.
Comment 7 Figures 5b and 5c are very nice. Did you use the same microscope for these images? If so, why are the resolutions of the other images, which are zoomed so poor?
Response. We are grateful for reviewer´s appreciation. Yes, we used the same microscope but different objectives. The images of this figures were obtained with 100x, whereas same other images were obtained with 40x or 60x and zoomed. We have changed some of those images by photographs taken with higher resolution (Figure 3B and 4B). In addition, the images of Höechst were taken from floating (detached) cells, not adhered to coverslips, which made harder to image them in high quality.
Comment 8 In Figure 6B. why not include the overlay? The yH2AX staining in Figure 6D, is not necessarily what is expected. In most yH2AX staining that I have seen, it is focal and solely in the nucleus (Oncogene volume23, pages2825–2837 (12 April 2004). Here, this looks like there is a lot of background/non-specific staining in the cytoplasm. How many cells were counted?
Response. We appreciate the suggestion made by the reviewer but we consider not necessary to add the overlay in figure 6B since it does not provide additional information.
We agree with the reviewer concerning the nuclear γH2AX staining. We have adjusted the contrast of images to better shown the nuclear expression of γH2AX (Figure 6D). For measuring γH2AX positive cells, 500 cells per condition, in three independent samples were counted. This has been indicated in the M&M section (lines 512-513).

Reviewer 2 Report
The authors show an interesting analysis on mytotic changes and alteration caused by the conversion of PPIX from MAL and its subsequent photoactivation. PDT therapy has been proposed already as alternative or coadjuvant strategy to treat tumors for a long time, but still is not in use on solid tumors or internal organs-deep tumor, where unfortunately the currently available laser/light sources equipment do not allow a deep penetration inside the human body.
Despite this observation, the authors show an impressive characterization of MTs alterations and mitotic changes which resemble the mechanism of action associated to drugs like nocodazole and other mitotic blockage drugs. However, the role of photosensiters in MTs aberrations has been already proven in this article “Cenklová, V. Photodynamic therapy with TMPyP - Porphyrine induces mitotic catastrophe and microtubule disorganization in HeLa and G361 cells, a comprehensive view of the action of the photosensitizer. J. Photochem. Photobiol. B. 2017, 173, 522-537”, reference 23 in the manuscript. Therefore in my opinion the authors should stress more about the novelty of their work, other than the usage of another PS.
Other comments for the authors:
1) The authors should contextualize the choice of HeLa cells tumor line. Is there a specific interest in treating carcinoma tumor? Otherwise if is just an example of tumor cell line the author should clearly state it and change the title of the article as well.
2) Figure 1: the conversion rate of the precursor MAL into the photoactive PPIX substrate has been characterize (Heme biosynthesis)and based on those data I should expect a decrease after few hours of the level of PPIX , especially considering the fast turnover of a tumor cell line. Also, it would be interesting to see the rate of photoconversion in the control cell line used as the non-tumorigenic counterpart (HaCaT), they should clearly see a difference in the conversion rate compared to HeLa cells.
3) Figure 2: is not convincing without a proper quantification, it looks like that the images were selected on purpose to match with the corresponding organelle staining, making difficult to judge if the colocalization come from technical procedure and crosstalk between he fluorescence signals or is a real colocalization.
4) Line 114-115 this statement is not clear to me, what the author are trying to explain?
5) Figure 3: the authors should add some control with the light alone or MAL without irradiation to confirm that the effect is coming from MAL+PDT combination, and exclude any possible effect coming from the light irradiation by itself which could potentially accumulate DNA damage in cells, even after 48 hours. The MTT assay shown in table 1 is not sensitive enough to detect DNA damages or MTs alterations.
Author Response
The authors show an interesting analysis on mytotic changes and alteration caused by the conversion of PPIX from MAL and its subsequent photoactivation. PDT therapy has been proposed already as alternative or coadjuvant strategy to treat tumors for a long time, but still is not in use on solid tumors or internal organs-deep tumor, where unfortunately the currently available laser/light sources equipment do not allow a deep penetration inside the human body.
Despite this observation, the authors show an impressive characterization of MTs alterations and mitotic changes which resemble the mechanism of action associated to drugs like nocodazole and other mitotic blockage drugs. However, the role of photosensiters in MTs aberrations has been already proven in this article “Cenklová, V. Photodynamic therapy with TMPyP - Porphyrine induces mitotic catastrophe and microtubule disorganization in HeLa and G361 cells, a comprehensive view of the action of the photosensitizer. J. Photochem. Photobiol. B. 2017, 173, 522-537”, reference 23 in the manuscript. Therefore in my opinion the authors should stress more about the novelty of their work, other than the usage of another PS.
Response. Some laboratories, including ours, have described the effects of other photosensitizers on MTs. Our work is focused on the prodrug MAL, which is one of the few compounds approved for cancer in the clinic (neither TMPyP nor other PSs that target MTs). Moreover, MAL-PDT is widely applied for the treatment of some types of non-melanoma skin cancer (in situ squamous cell carcinoma, superficial basal cell carcinoma), which represents the most common cancer type. It is also used for the treatment of the precancerous lesion, actinic keratosis. We consider that the study of the action mechanisms underlying its effectiveness could contribute to optimize this technology and might help to understand lack of response or recurrence after the treatment.
Comment 1 The authors should contextualize the choice of HeLa cells tumor line. Is there a specific interest in treating carcinoma tumor? Otherwise if is just an example of tumor cell line the author should clearly state it and change the title of the article as well.
Response. HeLa human carcinoma cells constitute a well-characterized and commonly used cell line for molecular and cellular studies in PDT. Thus, we employed this cell line to better understand the action mechanisms of MAL-PDT and so, the obtained results could be used as in vitro reference in laboratories focused on PDT action. According with the comment of the referee, we have included in the MS the reason why we have chosen this cell line (M&M section, lines 391-392). We have also changed the title of the article according with the suggestion of the reviewer “Mitotic catastrophe induced in tumor cells by Photodynamic therapy with methyl-aminolevulinate” to “Mitotic catastrophe induced in HeLa tumor cells by Photodynamic therapy with methyl-aminolevulinate”.
Comment 2 Figure 1: the conversion rate of the precursor MAL into the photoactive PPIX substrate has been characterize (Heme biosynthesis) and based on those data I should expect a decrease after few hours of the level of PPIX, especially considering the fast turnover of a tumor cell line. Also, it would be interesting to see the rate of photoconversion in the control cell line used as the non-tumorigenic counterpart (HaCaT), they should clearly see a difference in the conversion rate compared to HeLa cells.
Response. We agree with they comment, and also the bibliography supports a decrease in the intracellular PpIX levels after some hours. However, our results using MAL in HeLa cells showed a consistent increase in PpIX production after 24 h incubation with MAL. We have added a supplementary figure including the production rate of PpIX in HaCaT cells in which not significant differences respect to the basal control PpIX was observed. We have changed the text accordingly (line 117-122): “The production of PpIX after 5 h of incubation with MAL resulted to be dependent on the MAL concentration (0.3 vs. 1 mM), whereas no significant differences were found due to the incubation times (5 vs. 24 h) at each MAL concentration (Figure 2D). In contrast, PpIX production in HaCaT cells was independent of both MAL concentrations and incubation times in all the experimental conditions tested (Supplementary Figure 1).”
Comment 3 Figure 2: is not convincing without a proper quantification, it looks like that the images were selected on purpose to match with the corresponding organelle staining, making difficult to judge if the colocalization come from technical procedure and crosstalk between he fluorescence signals or is a real colocalization.
Response. According to the comment made by the referee, we have used FIJI to calculate the percentage of co-localization (PpIX and fluorescent markers) (100 cells were measured with the corresponding exciting lights). For that, we first established, separately, the threshold for positive cells in PpIX and in the organelle marker fluorescent images, obtained under the corresponding exiting light: UV (360–390 nm), blue (450–490 nm) and green (510–550 nm). The positive area for PpIX as well as that shared with the fluorescent the organelle marker was measured. Then, the percentage of shared positivity with respect to the organelle marker was calculated. Two representative images are shown in the Word file. We have included a diagram in figure 2B’ to show the obtained results confirming that PpIX is mainly localized in HeLa lysosomes.
Comment 4 Line 114-115 this statement is not clear to me, what the author are trying to explain?
Response. We have modified the sentence as follows: “Since we detected changes in the cellular response to PDT when using different treatment conditions, we analyzed by flow cytometry the levels of PpIX produced in HeLa cells (Figure 2C).”
Comment 5 Figure 3: the authors should add some control with the light alone or MAL without irradiation to confirm that the effect is coming from MAL+PDT combination, and exclude any possible effect coming from the light irradiation by itself which could potentially accumulate DNA damage in cells, even after 48 hours. The MTT assay shown in table 1 is not sensitive enough to detect DNA damages or MTs alterations.
Response. According with the comment made by the referee, we have included data showing the nuclear morphology as well as the expression of γH2Ax to determine DNA integrity in cells subjected to MAL or light alone. The data indicate that cells treated with MAL or light alone, showed an expression of γH2Ax similar to controls, differing from that observed in MAL-PDT treated cells (Supplementary Figure 2). In agreement with our results, it has been previously published that irradiation with light or incubation with other PpIX precursor (5-ALA) did not produce DNA damage (Abo-Zeid, M. A., Abo-Elfadl, M. T., & Mostafa, S. M. Photodynamic therapy using 5-aminolevulinic acid triggered DNA damage of adenocarcinoma breast cancer and hepatocellular carcinoma cell lines. Photodiagnosis and photodynamic therapy, 2018, 21, 351-356).

Round 2
Reviewer 1 Report
The authors have made the appropriate changes.
Reviewer 2 Report
The authors have addressed all the concerns and questions raised. I'm satisfied.